# KCTD19 and its associated protein ZFP541 are independently essential for meiosis in male mice

Seiya Oura[1,2], Takayuki Koyano[3], Chisato Kodera[4], Yuki Horisawa-Takada[4], Makoto Matsuyama[3], Kei-ichiro Ishiguro[4], Masahito Ikawa[1,2,5] *

1 Research Institute for Microbial Diseases, Osaka University, Osaka, Japan, 2 Graduate School of Pharmaceutical Sciences, Osaka University, Osaka, Japan, 3 Division of Molecular Genetics, Shigei Medical Research Institute, Okayama, Japan, 4 Department of Chromosome Biology, Institute of Molecular Embryology and Genetics (IMEG), Kumamoto University, Kumamoto, Japan, 5 The Institute of Medical Science, The University of Tokyo, Minato-ku, Tokyo, Japan

* ikawa@biken.osaka-u.ac.jp

**Data Availability Statement:** All relevant data are within the manuscript and its Supporting Information files.

## Abstract

Meiosis is a cell division process with complex chromosome events where various molecules must work in tandem. To find meiosis-related genes, we screened evolutionarily conserved and reproductive tract-enriched genes using the CRISPR/Cas9 system and identified potassium channel tetramerization domain containing 19 (*Kctd19*) as an essential factor for meiosis. In prophase I, *Kctd19* deficiency did not affect synapsis or the DNA damage response, and chiasma structures were also observed in metaphase I spermatocytes of *Kctd19* KO mice. However, spermatocytes underwent apoptotic elimination during the metaphase-anaphase transition. We were able to rescue the *Kctd19* KO phenotype with an epitope-tagged *Kctd19* transgene. By immunoprecipitation-mass spectrometry, we confirmed the association of KCTD19 with zinc finger protein 541 (ZFP541) and histone deacetylase 1 (HDAC1). Phenotyping of *Zfp541* KO spermatocytes demonstrated XY chromosome asynapsis and recurrent DNA damage in the late pachytene stage, leading to apoptosis. In summary, our study reveals that KCTD19 associates with ZFP541 and HDAC1, and that both KCTD19 and ZFP541 are essential for meiosis in male mice.

## Author summary

Meiosis is a fundamental process that consists of one round of genomic DNA replication and two rounds of chromosome segregation, producing four haploid cells. To properly distribute their genetic material, cells need to undergo complex chromosome events such as a physical linkage of homologous chromosomes (termed synapsis) and meiotic recombination. The molecules involved in these events have not been fully characterized yet, especially in mammals. Using a CRISPR/Cas9-screening system, we identified the potassium channel tetramerization domain containing 19 (*Kctd19*) as an essential factor for meiosis in male mice. Further, we confirmed the association of KCTD19 with zinc finger protein 541 (ZFP541) and histone deacetylase 1 (HDAC1). By observing meiosis of

**Funding:** This work was supported by Ministry of Education, Culture, Sports, Science and Technology (MEXT)/Japan Society for the Promotion of Science (JSPS) KAKENHI grants (JP19J21619 to S.O., 19H05743 to K.I., and JP19H05750 to M.I.); Japan Agency for Medical Research and Development (AMED) grant JP20gm5010001 to M.I.; Takeda Science Foundation grants to M.I. The funders had no role in the study design, data collection and analysis, decision to publish, or preparation of the manuscript.

**Competing interests:** The authors have declared that no competing interests exist.

*Zfp541* knockout germ cells, we found that *Zfp541* was also essential for meiosis. These results show that the KCTD19/ZFP541 complex plays a critical role and is indispensable for male meiosis and fertility.

## Introduction

Meiosis is a division process consisting of one round of DNA replication and two rounds of chromosome segregation, producing four haploid gametes. During meiotic prophase I, proteinaceous structures termed the synaptonemal complex (SC) are assembled on sister chromatids and form a scaffold along each homologous chromosome. The homologs begin to pair and synapse, followed by meiotic recombination yielding a physical tether between homologs (chiasmata). After completing these chromosome events, the cells transition to the first meiotic division, where homologs are segregated to the opposite poles, followed by the segregation of sister chromatids in the next round of cell division.

The molecules involved in these complex chromosome events are not fully characterized yet, especially in mammals, due to difficulties in culturing and genetically manipulating spermatogenic cells *in vitro*. Thus, knockout (KO) of genes with testis-specific expression and evolutionarily conservation has been a powerful strategy to identify male meiosis-related genes and their functions [1]. We have generated over 300 testis-enriched gene KO mice with conventional ES cell-mediated and the CRISPR/Cas9-mediated methods [2–5] and showed about one-third of them are indispensable for male fertility [6–8]. During this phenotypic screening, we identified potassium channel tetramerization domain containing 19 (*Kctd19*) as an evolutionarily conserved and testis expressed gene that is essential for male fertility in mice.

KCTD19 is one of a 26 member KCTD family of proteins [9,10] (KCTD1–21, KCTD12B, TNFAIP1, KCNRG, SHKBP1, and BTBD10) which contains an N-terminal cytoplasmic tetramerization domain (T1) usually found in voltage-gated potassium channels. The T1 domain is a subgroup of the BTB (Broad-complex, Tramtrack and Bric-à-brac) or POZ (poxvirus and zinc finger) domain family, often found at the N-terminus of C2H2-type zinc-finger transcription factors. A variety of biological functions have been identified for KCTD proteins [10], including ion channel regulation [11,12], apoptosis [13,14], interaction with ubiquitin ligase complexes such as cullin 3 (CUL3) [15,16], and degradation of various proteins such as histone deacetylases (HDACs) [15,17]. Regarding KCTD19, Choi *et al.* found that ZFP541 complexes with KCTD19 and HDAC1 in male germ cells and valproic acid (HDAC inhibitor) treatment caused hyperacetylation and KCTD19/ZFP541 reduction in round spermatids [18], suggesting that KCTD19/ZFP541 are involved in chromatin reorganization during the post-meiotic phase [18].

In this study, we generated *Kctd19* KO mice using the CRISPR/Cas9 system and revealed that *Kctd19* deficiency causes azoospermia due to incomplete meiosis. Then, we confirmed KCTD19, ZFP541, and HDAC1 interaction by immunoprecipitation-mass spectrometry (IP-MS). Further, we also analyzed *Zfp541* null spermatocyte and showed that *Zfp541* is necessary for pachytene exit. Our results showed that a KCTD19/ZFP541 complex functions during male meiosis.

## Results

### *Kctd19* is a testis-enriched and evolutionarily conserved gene

To investigate the spatial expression of *Kctd19* in mice, we performed multi-tissue RT-PCR using cDNA obtained from adult tissues and embryonic ovary, and we found that *Kctd19* was

specifically expressed in testis (Fig 1A). In mice, the first wave of spermatogenesis starts soon after birth and completes within the first 35 days of postnatal development [19]. To determine which stage of spermatogenic cells begin to express *Kctd19*, we also performed RT-PCR using cDNA obtained from postnatal testis as the first wave of spermatogenesis was progressing. The result shows that *Kctd19* expression starts around postnatal day (PND) 10–12 (Fig 1B), which corresponds to the spermatocyte stage when the first wave of spermatogenesis reaches meiotic prophase. The PCR signals increased until PND 28 (Fig 1B), at which time spermatid elongation starts.

The mouse KCTD19 protein comprises 950 amino-acid residues and has only one BTB domain based on SMART software [20] (Fig 1C). Phylogenetic analysis with Clustal W2.1 [21] showed that KCTD19 was evolutionarily conserved in many mammals, including cattle, dogs, mice, and humans (Figs 1D and S1). These results suggest that KCTD19 functions during the meiotic phase of mammalian spermatogenesis.

## *Kctd19* is required for male fertility

To uncover the function of *Kctd19* in vivo, we generated *Kctd19* KO mice using the CRISPR/Cas9 system. To avoid affecting the proximal genes, *Lrrc36* and *Plekhg4*, we designed the excision of the middle exons 3–12 from 16 exons total (Fig 1E). Two crRNAs were mixed with tracrRNA and Cas9, and the prepared ribonucleoproteins (RNPs) were electroporated into murine zygotes. Of the 49 fertilized eggs that were electroporated, 40 two-cell embryos were transplanted into the oviducts of three pseudopregnant female mice. We obtained seven pups with the intended mutation. Subsequent mating and sequencing resulted in a heterozygous mouse with a 9620 bp deletion, referred to as *Kctd19*$^{del}$ that were genotyped with PCR (Fig 1F and 1G). We confirmed *Kctd19* deletion with immunoblotting (Fig 1H) with various antibodies raised against KCTD19 protein (see Fig 1C). The results showed complete loss of KCTD19 in *Kctd19*$^{del/del}$ testis, and validated the antibodies recognize KCTD19 (Fig 1H). We used rabbit polyclonal antibody (pAb) and rat monoclonal antibody (mAb) #19–3 for immunoprecipitation and rat mAb #22–15 for immunostaining in subsequent experiments.

Knockout (KO) mice obtained by heterozygous intercrosses showed no overt gross defects in development, behavior, and survival. We caged individual *Kctd19*$^{del/del}$ male mice with wild type (wt) females for two months to analyze their fertility. Although mating plugs were often observed, *Kctd19*$^{del/del}$ males failed to sire any pups (Fig 1I). We observed normal numbers of pups from *Kctd19*$^{del/del}$ females with *Kctd19*$^{wt/del}$ males (7.8±2.2; Fig 1J), indicating that *Kctd19* is not required for female fertility. As *Kctd19*$^{wt/del}$ male mice are fully fertile, we used littermate heterozygous males as controls in some experiments.

To determine if the BTB domain of KCTD19 is required for protein function, we removed exon 2 (297 bp) that encodes the BTB domain by designing two crRNAs targeting intron 1 and 2 (S2A–S2C Fig). Despite generating an inframe mutation, the deletion of the BTB domain affected *Kctd19* expression or/and protein stability, and we could not detect any truncated KCTD19 protein with our antibodies (S2D and S2E Fig). The exon 2 deleted mice showed the same phenotype as *Kctd19*$^{del/del}$ mice (Figs S2F and 2C). Therefore, we regarded this *Kctd19*-ΔBTB line as equivalent to *Kctd19*$^{del/del}$ line, in that both lines result in male infertility, to corroborate that *Kctd19* is essential for male fertility.

## *Kctd19*$^{del/del}$ spermatocytes failed to complete meiosis

When we observed testis gross morphology, *Kctd19*$^{del/del}$ testis were smaller than those of *Kctd19*$^{wt/del}$ (testis/ body weight: 4.5±0.2 x 10$^{-3}$ [*wt/del*], 1.2±0.3 x 10$^{-3}$ [*del/del*]; Fig 2A and 2B), indicating defective spermatogenesis in *Kctd19*$^{del/del}$ testis. To define the cause of testicular

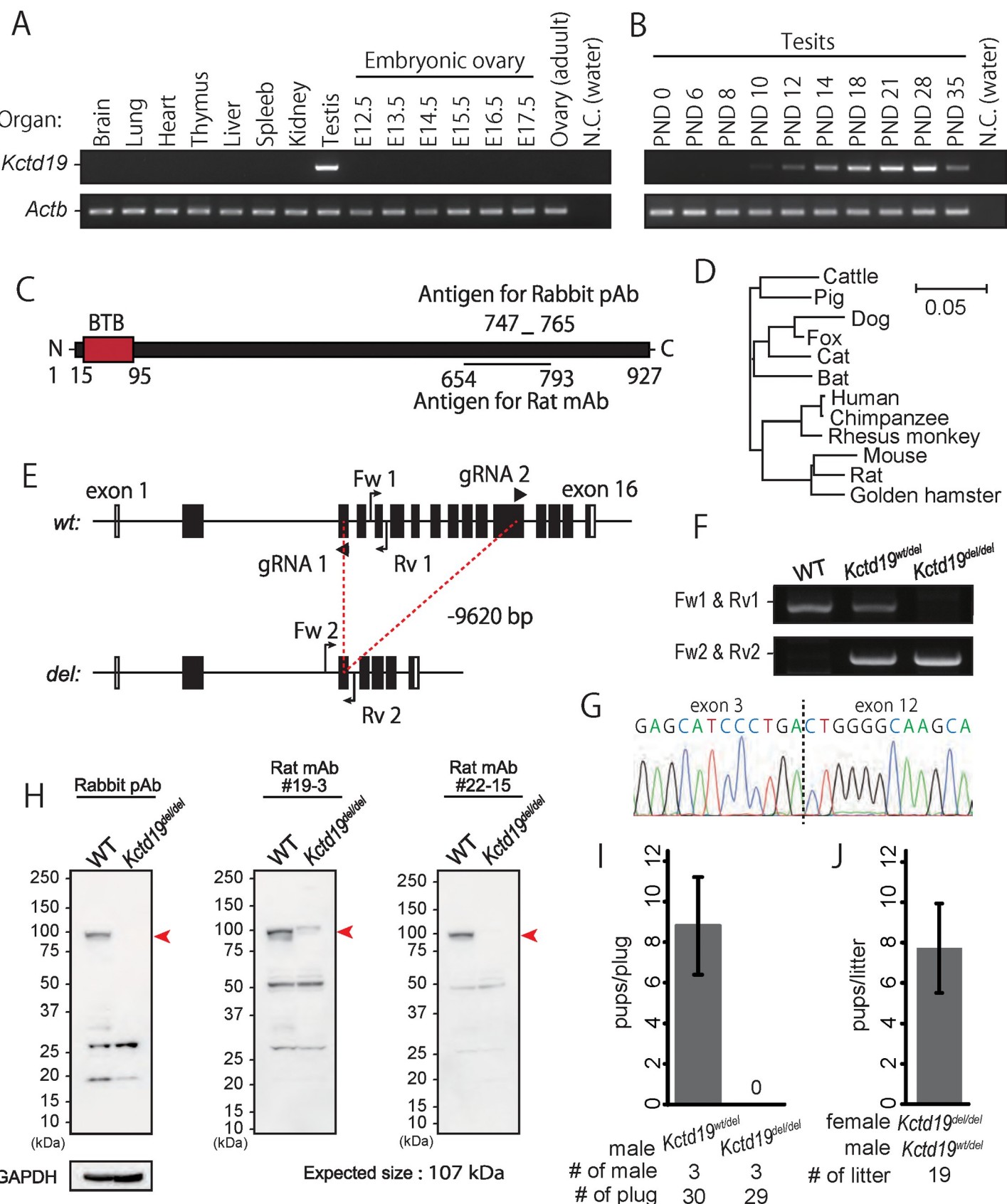

**Fig 1. Production of *Kctd19*$^{del/del}$ mice and fertility analysis.** (A) RT-PCR using multi-tissue cDNA. *Actb* was used as a loading control. (B) RT-PCR using postnatal testis cDNA. *Actb* was used as a loading control. (C) Schematic of KCTD19 protein structure and antigen position. (D) Phylogenetic tree constructed by ClustalW with KCTD19 sequences of various mammals. (E) Gene map of *Kctd19*. Black and white boxes indicate coding and non-coding regions, respectively. Black arrows and arrowheads indicate primers for genotyping and gRNAs for genome editing, respectively. (F) An example of genotyping PCR with two primer sets shown in E. (G) DNA sequencing for deletion verification. (H) Immunoblotting with antibodies against mouse KCTD19. Red arrows indicate the expected molecular size of KCTD19. GAPDH was used as a loading control. (I) The result of mating tests. Pups/plug: 8.8±2.4 [WT]; 0 [*del/del*]. (J) Pup numbers obtained from mating pairs of *Kctd19*$^{del/del}$ females and *Kctd19*$^{wt/del}$ males (7.8±2.2).

atrophy, we performed hematoxylin and periodic acid-Schiff (HePAS) staining of testicular sections. While three germ cell layers were seen in control testis sections, only two layers of germ cells were observed in *Kctd19*$^{del/del}$ testis (Fig 2C; low magnification). When we compared testicular cells based on the cycle of the seminiferous epithelium [22–23], the nuclear morphology of spermatocytes was comparable between the two genotypes up to seminiferous stage X–XI, corresponding to the diplotene stage (Fig 2C). In seminiferous stage XII, spermatocytes proceeded to metaphase-anaphase in *Kctd19*$^{del/del}$ testis as well as in *Kctd19*$^{wtl/del}$ testis (Fig 2C). However, the *Kctd19*$^{del/del}$ spermatocytes could not complete meiotic divisions and accumulated in tubules after stage XII (Fig 2C; stage I–II). These accumulated spermatocytes underwent apoptosis (Fig 2D and 2E) and did not develop to haploid spermatids. As a result, no mature spermatozoa were observed in the cauda epididymis (Fig 2F). These observations suggested that *Kctd19*$^{del/del}$ spermatocytes failed to complete meiosis, leading to azoospermia.

## KCTD19 localized to the nuclei of prophase spermatocytes and round spermatids

To determine KCTD19 localization, we performed immunostaining of testicular sections with a specific antibody against KCTD19 (Rat mAb #22–15; Fig 2G). KCTD19 signals started to appear in the nuclei of spermatocytes in seminiferous stage III–IV (Fig 2H), corresponding to early pachytene stage. The signal continuously localized in the nuclei of spermatocytes (Fig 2H; stage VII–VIII and X–XI). During the metaphase-anaphase transition in meiosis, KCTD19 signal spread throughout the cell (Fig 2H; stage XII). The signals remained in the nuclei of round spermatids after meiotic division and disappeared in elongating spermatids. The KO phenotype and KCTD19 localization suggested that KCTD19 regulates meiosis in spermatocyte nuclei.

## *Kctd19*$^{del/del}$ spermatocytes showed defects in metaphase I organization

Due to an apparent defect in meiosis in *Kctd19*$^{del/del}$ male mice, we examined DNA double-strand breaks (DSBs) and synapsis by immunostaining γH2AX and synaptonemal complex protein 3 (SYCP3), respectively. γH2AX signals appeared in the leptotene/zygotene stage and disappeared in the pachytene/diplotene stage, except for the XY body (Figs 3A and S3A), suggesting that *Kctd19*$^{del/del}$ spermatocytes underwent DSB initiation and resolution as controls. Also, homologous chromosomes in *Kctd19*$^{del/del}$ spermatocytes synapsed in pachytene stage and desynapsed in diplotene stage without obvious defects (Fig 3A and S3A). However, the diplotene population declined in juvenile *Kctd19*$^{del/del}$ males (P20; S3B Fig), but not in adult males (Fig 3B)

To uncover the cause of apoptosis in metaphase spermatocytes, we stained spread chromosomes with Giemsa staining. We observed a normal number of bivalent chromosomes with chiasmata (Fig 3C). Next, we examined spindles in metaphase I spermatocytes by immunostaining of CENPC and α-TUBULIN. Although *Kctd19*$^{del/del}$ spermatocytes formed spindles without apparent defects, they showed chromosome misalignment (Fig 3D and 3E; WT: 0%,

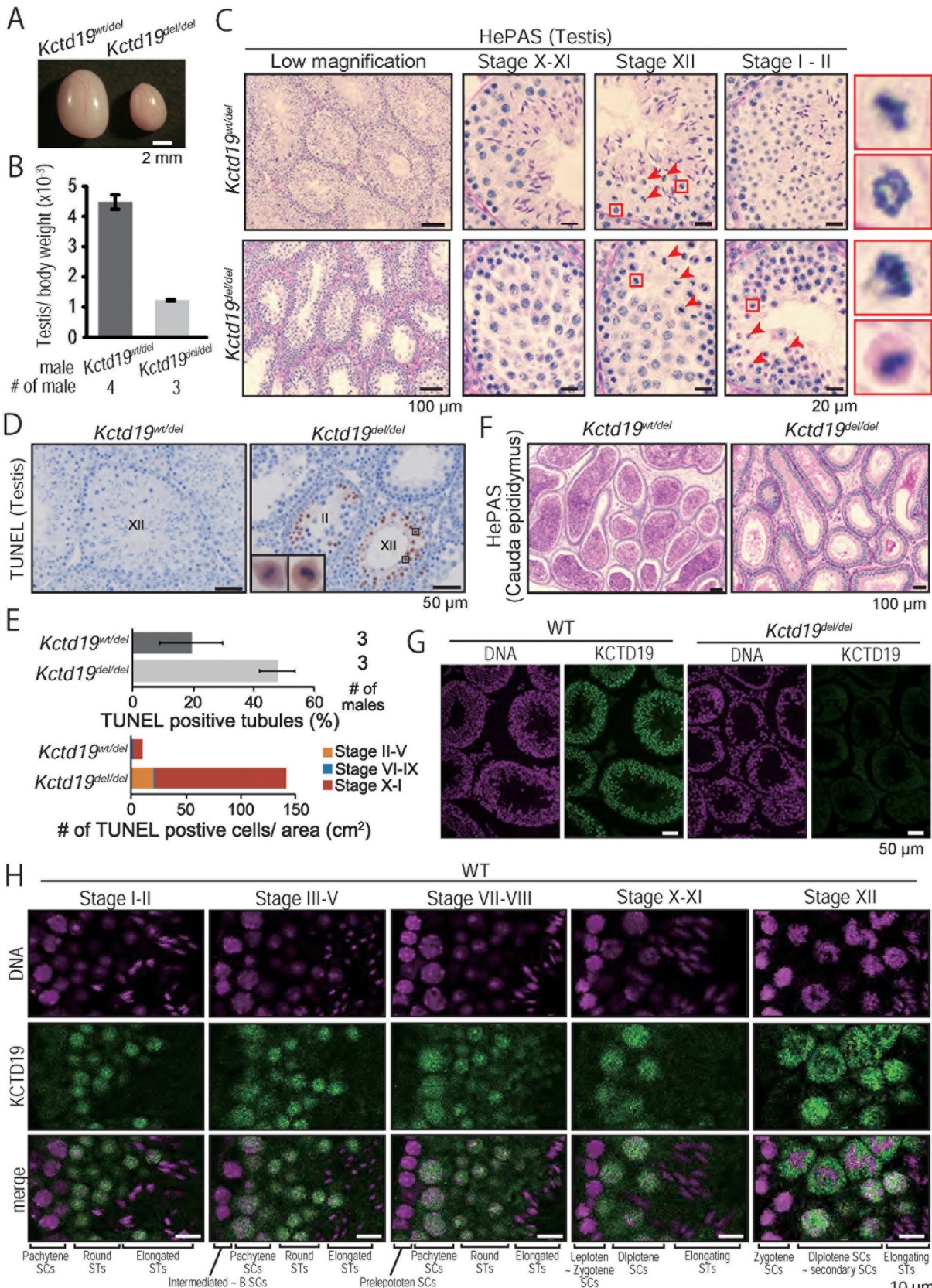

**Fig 2. Histological analysis of *Kctd19*<sup>del/del</sup> mice.** (A) Testis morphology and (B) testis/body weight of *Kctd19*<sup>wt/del</sup> and *Kctd19*<sup>del/del</sup> adult mice at 12 weeks. Testis/body weight: $4.5\pm0.2 \times 10^{-3}$ [*wt/del*], $1.2\pm0.3 \times 10^{-3}$ [*del/del*]. Error bars indicate one standard deviation. (C) PAS staining of seminiferous tubules of adult mice. The seminiferous epithelium cycle was determined by germ cell position and nuclear morphology. Red arrows and boxes indicate metaphase cells. (D) TUNEL staining of seminiferous tubules of adult mice counterstained with hematoxylin. (E) The seminiferous epithelial stages were roughly determined by the arrangement and nuclear morphology of the first layer of germ cells (spermatogonia and leptotene/zygotene spermatocytes). (F) PAS staining of cauda epididymis of adult mice. (G & H) Immunostaining of seminiferous tubules of adult mice. The seminiferous epithelium cycle was determined by cell position, nuclear morphology, and morphology of acrosome staining with Alexa Flour 568-conjugated lectin PNA.

*del/del*: 33%). When we stained SYCP3, we observed SYCP3 aggregates outside chromosomes, known as polycomplexes [24], more frequently in *Kctd19*<sup>del/del</sup> than in WT metaphase spermatocytes (WT: 12%, *del/del*: 65%; Fig 3F and 3G). These results suggested that KCTD19 is required for metaphase I organization.

## An epitope-tagged transgene rescues the phenotype of *Kctd19*<sup>del/del</sup> mice

To exclude the possibility that the observed phenotype in *Kctd19*<sup>del/del</sup> males was caused by an off-target effect from CRISPR/Cas9 cleavage or an aberrant genetic modification near the *Kctd19* locus, we carried out a rescue experiment by generating transgenic (Tg) mouse lines. We mixed and injected two DNA constructs having 3xFLAG-tagged *Kctd19* and 3xHA-tagged *Kctd19* under the testis-specific *Clgn* promoter [25] (Fig 4A) and established two Tg lines: one expressing only 3xHA-tagged *Kctd19* (Tg line #1 [HA]; Fig 4B and 4C) and one expressing both 3xFLAG- and 3xHA-tagged *Kctd19* (Tg line #2 [HA/FLAG]; Fig 4B and 4D). When we performed immunoprecipitation (IP) with Tg line #2 [HA/FLAG], anti-FLAG antibody-conjugated beads pull downed 3xHA-KCTD19, and vice versa (Fig 4E and 4F), suggesting that KCTD19 is a homomeric protein as previously reported [9,26].

When we mated Tg positive *Kctd19*<sup>del/del</sup> male mice with superovulated WT female mice (Fig 4G), we could obtain 2-cell embryos from both Tg lines (Fig 4H and 4I), showing recovery of fertility. In *Kctd19*<sup>del/del</sup> mice carrying the 3xHA-*KCTD19* transgene (Tg line #1 [HA]), the testicular size (testis/body weight: $4.7 \pm 1.6 \times 10^{-3}$; Fig 4J and 4K) was comparable to WT, and spermatogenesis evaluated by HePAS staining looked normal (Fig 4L). Further, with an anti-HA antibody, we observed a similar immunostaining pattern with rat monoclonal anti-KCTD19 (Fig 2G), indicating that the 3xHA-tag did not affect KCTD19 behavior and corroborating the immunostaining results of the anti-KCTD19 antibody.

## KCTD19 associates with ZFP541 and HDAC1

To elucidate KCTD19 function, we identified interacting proteins by immunoprecipitation (IP) and mass spectrometry (MS). We lysed *Kctd19*<sup>del/del</sup> and juvenile (PND21) WT testis with non-ionic detergent (NP40) and incubated the lysate with antibodies (rabbit pAb and rat mAb #1) and protein G-conjugate beads. The specific co-IPed proteins were visualized by SDS-PAGE and silver staining (Fig 5A and 5B). We used both the rabbit polyclonal and rat monoclonal (#19–3) antibodies for narrowing the interacting candidates. When eluted samples were subjected to MS analysis, HDAC1 (histone deacetylase 1) and ZNF541 (Zinc finger protein 54; ZFP541) were reproducibly detected with both antibody IPs (Fig 5C), consistent with a prior study [18]. KCTD19 and HDAC1 association was confirmed by reciprocal IP with an anti-HDAC1 antibody (Fig 5D).

HDAC1 is a modulator of chromatin structure and disruption of HDAC1 results in embryonic lethality before E10.5 [27] In previous reports, KCTDs were implicated in HDAC degradation (15, 17). We examine the behavior of HDAC1 in *Kctd19*<sup>del/del</sup> testis by immunoblotting analysis and immunostaining with the anti-HDAC1 antibody. HDAC1 protein levels and

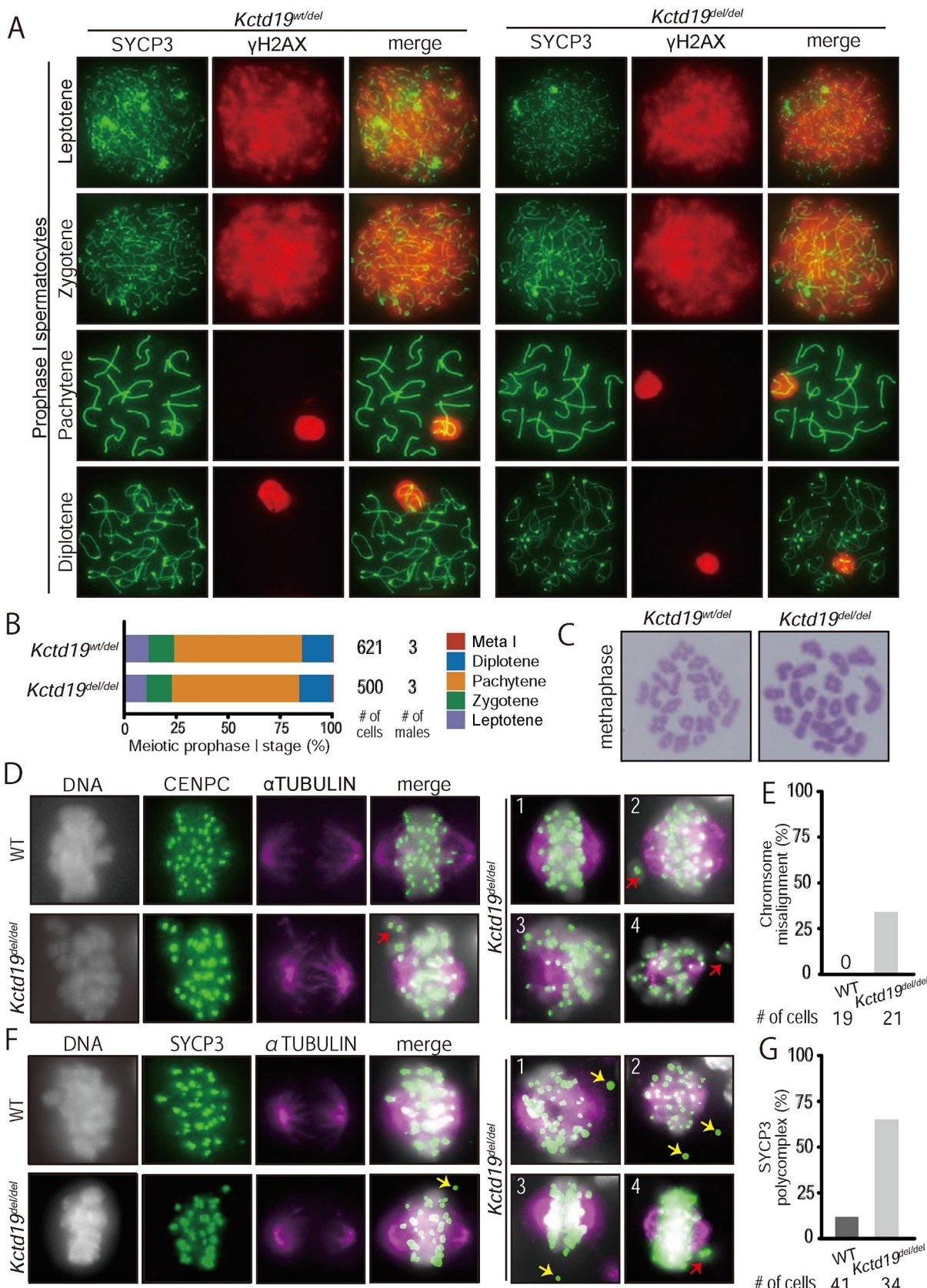

**Fig 3. Cytological analysis of *Kctd19^{del/del}* spermatocytes.** (A) Immunostaining of spread nuclei from prophase spermatocytes collected from adult mice. (B) The percentage of each meiotic prophase stage present is determined by immunostained spread nuclei samples. (C) Giemsa staining of spread nuclei of metaphase I spermatocytes. (D) Immunostaining of prophase spermatocytes with antibodies against CENPC and α-TUBULIN. Right panels (1–4) show additional *Kctd19^{del/del}* spermatocytes. Red arrows indicate misaligned chromosomes. (E) The percentage of metaphase I spermatocytes with misaligned chromosomes. (F) Immunostaining of prophase spermatocytes with antibodies against SYCP3 and α-TUBULIN. Right panels (1–4) show additional *Kctd19^{del/del}* spermatocytes. Red and yellow arrows indicate misaligned chromosomes and SYCP3 polycomplexes, respectively. (G) The percentage of metaphase I spermatocytes with SYCP3 polycomplexes.

localization were comparable between *Kctd19^{del/del}* and WT testis (Fig 5E and 5F). In WT testis, the HDAC1 staining intensity became the strongest in stage X–XI spermatocytes and lost in elongating spermatids (Fig 5G), reminiscent of the KCTD19 staining pattern (Fig 2G). These results indicated that KCTD19 works together with HDAC1 in regulating meiosis.

## *Zfp541* deficient spermatocytes fail to exit the pachytene stage

The second factor identified by co-IP MS analysis, *Zfp541*, is specifically expressed in testis (Fig 6A). Further, *Zfp541* expression began around PND10–12 and was then continuously detected with increasing signal intensity at PND28 (Fig 6B), reminiscent of *Kctd19* rtPCR (Fig 1B). The mouse ZFP541 protein comprises 1363 amino-acid residues and has five C2H2 type zinc finger motifs, one ELM2 domain, and one SANT domain based on SMART software [20] (Fig 6C), indicating the KCTD19/ZFP541 complex may bind DNA. Phylogenetic analysis with Clustal W2.1 [21] showed that *Zfp541* was also evolutionarily conserved in many mammals (Figs 6D and S4). To reveal the function of ZFP541 and its relationship with KCTD19, we analyzed *Zfp541* KO phenotype with chimeric mice (chimeric analysis), which enables rapid gene functional analysis [4–5].

To disrupt gene function completely and minimize an effect on the juxtaposed gene, *Napa*, we designed two sgRNAs targeting the sequence upstream of the start codon and intron 8 (Fig 6E), and transfected embryonic stem (ES) cells expressing EGFP [28] with two pairs of sgRNA/Cas9 expressing plasmids (pair-1: gRNA 1 and 3; pair-2: gRNA 2 and 4; Fig 6E). We Screened 32 clones for each pair, and obtained 13 and 11 mutant clones with biallelic deletion for pair-1 and -2. Accounting for ES cell quality and off-target cleavages, we produced chimeric mice with one ES cell clone from pair-1 (1–3 #2) and pair-2 (2–4 #3) (Fig 6F and 6G).

First, we examined spermatogenesis with HePAS staining of testicular sections. Almost no round spermatids were observed in GFP positive seminiferous tubules of chimeric mice (Fig 6H), as seen in *Kctd19^{del/del}* testis sections. *Zfp541* deficient spermatocytes were mainly eliminated by apoptosis in stage X–I seminiferous tubules (Fig 6I and 6J). Next, we performed immunostaining with the antibodies against KCTD19. The KCTD19 intensity became weaker, although not lost, in the nuclei of *Zfp541* deficient spermatocytes than that of adjacent WT spermatocytes (Fig 7A). On the other hand, the immunofluorescence intensity of HDAC1 was comparable between *Zfp541* deficient and WT spermatocytes (Fig 7B). Finally, we examined the DNA damage response and synapsis in a XX/XY (Host/ES) chimeric male mouse [29], in which all spermatocytes are derived from the mutant ES cells (S5A and S5B Fig). *Zfp541* deficient spermatocytes initiated DSBs in the leptotene/zygotene stage and resolved the breaks in the early pachytene stage (Fig 7C). However, late pachytene spermatocytes showed recurrent DSBs. When we meticulously examined early pachytene spermatocytes, we could observe asynapsis of XY chromosomes (red and yellow boxes in Fig 7C). Further, SYCP1 spread outside of the synaptonemal complex axis in the late pachytene stage (Fig 7D). No diplotene spermatocytes were observed in the chimeric mouse (Fig 7E). Collectively, these results showed that *Zfp541* deficient spermatocytes did not reach the diplotene stage. Thus, KCTD19 may function downstream of ZFP541.

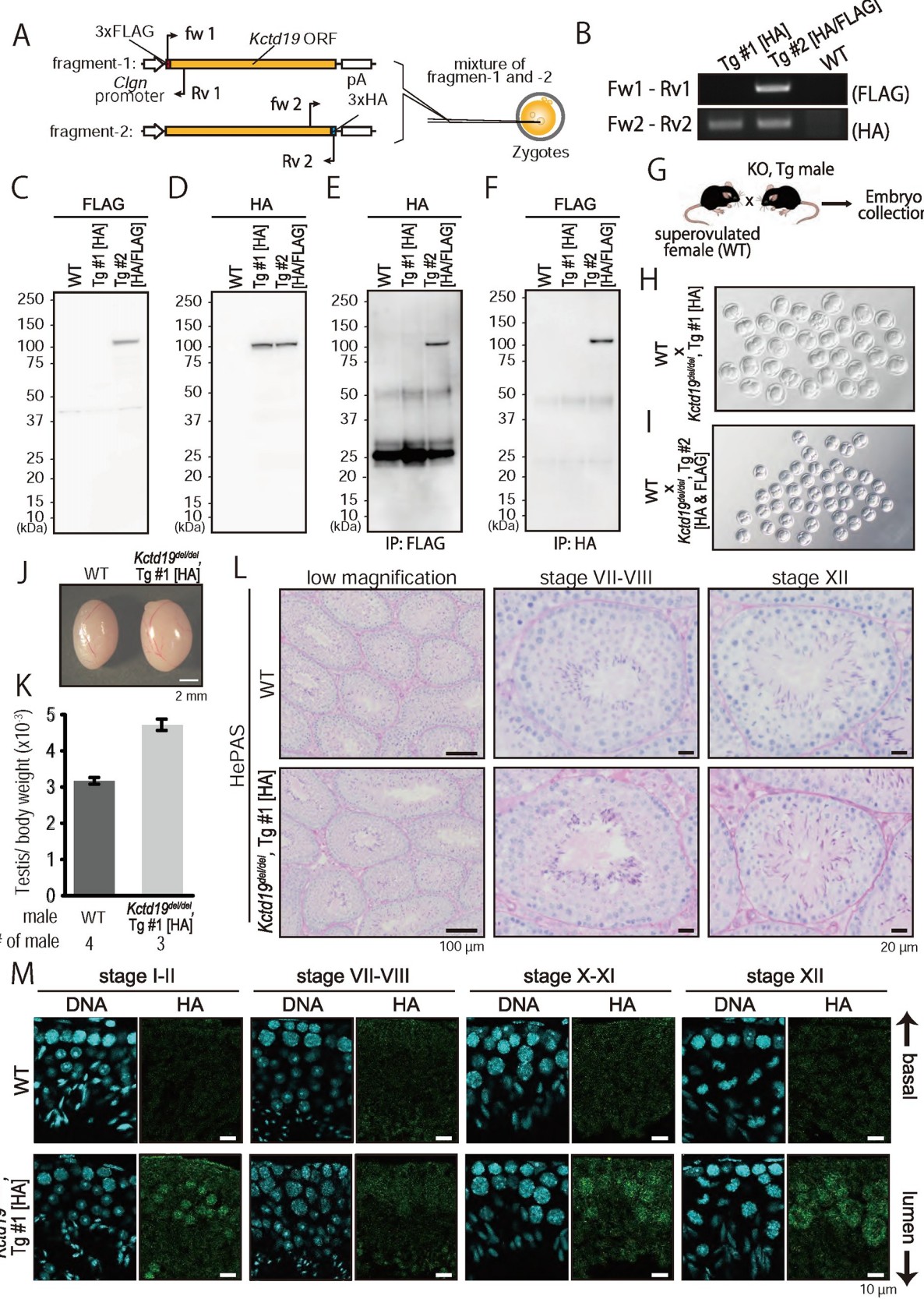

**Fig 4. Transgenic (Tg) rescue of *Kctd19^del/del* mice.** (A) Schematic of Tg mouse production. Red and yellow boxes indicate affinity tags and *Kctd19* open reading frame, respectively. Black arrows indicate primers for genotyping. (B) An example of PCR genotyping with two primer sets shown in A. (C & D) Immunoblotting with antibodies against FLAG (C) and HA (D) for determining expression levels. (E & F) Immunoprecipitation with antibodies against FLAG (E) and HA (F) and immunoblotting with antibodies against HA and FLAG, respectively. (G) Schematics of fertility determination of Tg mice. (H & I) Two-cell embryos obtained from WT females mated with *Kctd19^del/del* males carrying transgenes. (J) Testis morphology and (K) testis/body weight of WT and *Kctd19^del/del*, Tg #1 adult mice at 8-week of age. Testis/body weight: $3.2 \pm 0.1 \times 10^{-3}$ [WT]; $4.7 \pm 1.6 \times 10^{-3}$ [*Kctd19^del/del*, Tg #1]. Error bars indicate one standard deviation. (L) PAS staining of seminiferous tubules of adult mice. The seminiferous epithelium cycle was determined by germ cell position and nuclear morphology. (M) Immunostaining of seminiferous tubules of adult mice. The seminiferous epithelium cycle was determined by cell position, nuclear morphology, and morphology of the acrosome stained with Alexa Flour 568-conjugated lectin PNA.

## Discussion

In the present study, we identified *Kctd19* as a male fertility-related factor by CRISPR/Cas9-mediated screening of testis enriched genes and validated our result with transgenic rescue experiments. Recently, Fang *et al.* also reported metaphase I arrest in *Kctd19* KO male mice [30], corroborating our results. In detailed phenotypic analyses, we found that *Kctd19* KO spermatocytes failed to complete meiotic division with defects in metaphase I organization. Further, we confirmed that KCTD19 associates with ZFP541 and HDAC1 by co-IP experiments using two antibodies against KCTD19. Finally, we produced chimeric mice with *Zfp541*-KO ES cells and showed that *Zfp541* is essential for pachytene exit.

*Kctd19* KO spermatocytes showed a metaphase-anaphase arrest and were eliminated by apoptosis. One of the most frequent causes of metaphase I arrest is crossover (CO) defects causing precocious homolog segregation [31–32]. However, *Kctd19* KO spermatocytes had a normal number of bivalents (20 homologs) in metaphase I, indicating that homologs were physically connected in *Kctd19* KO spermatocytes. We also observed SYCP3 polycomplexes [24], alternative SC structures, in metaphase I spermatocytes. A common cause of synaptonemal polycomplex formation is an excess amount of free SC components [24], which might be caused by premature dissociation of SC or misregulation of SC-related protein expression. However, we could not rule out the possibility that these metaphase I structural defects might be a secondary effect or phenomena in dying cells. In addition, we observed a delay of metaphase entry or elimination during prophase I in juvenile *Kctd19* KO males (PND20), indicating that KCTD19 may function during prophase or that the first wave of spermatogenesis is exceptional.

To clarify the molecular function of KCTD19, we tried to identify interacting proteins by IP-MS analysis and found ZFP541 and HDAC1 as candidate proteins, consistent with the previous report [18]. Although some KCTD members have been reported to be associated with HDAC degradation, we could not observe changes in HDAC1 amount in *Kctd19* KO testis by immunoblotting or immunostaining analysis. On the other hand, the KCTD19/ZFP541 complex is reminiscent of BTB-ZF proteins, which have another subset of the BTB domain and the Krüppel-type C2H2 zinc fingers [33]. Many BTB-ZF proteins have been implicated in transcriptional repressorion such as N-CoR, SMRT, and HDACs via the BTB domain [34–36]. The ELM2-SANT domain included in ZFP541 has also been shown to interact with HDAC1 [18,37–39]. Combined with these previous reports, our results suggested that the KCTD19/ZFP541 complex works on chromatin modification of spermatocytes with HDAC1. We also detected CUL9 and DNTTIP1 in the IP-MS analysis with rabbit-generated anti-KCTD19 antibody, albeit not with the rat mAb #19–3. These factors can be excellent targets in future research because knockdown experiments from other groups showed that CUL9 protects mouse eggs from aneuploidy [40] and DNTTIP1 loss causes chromosome misalignment in mitosis [41].

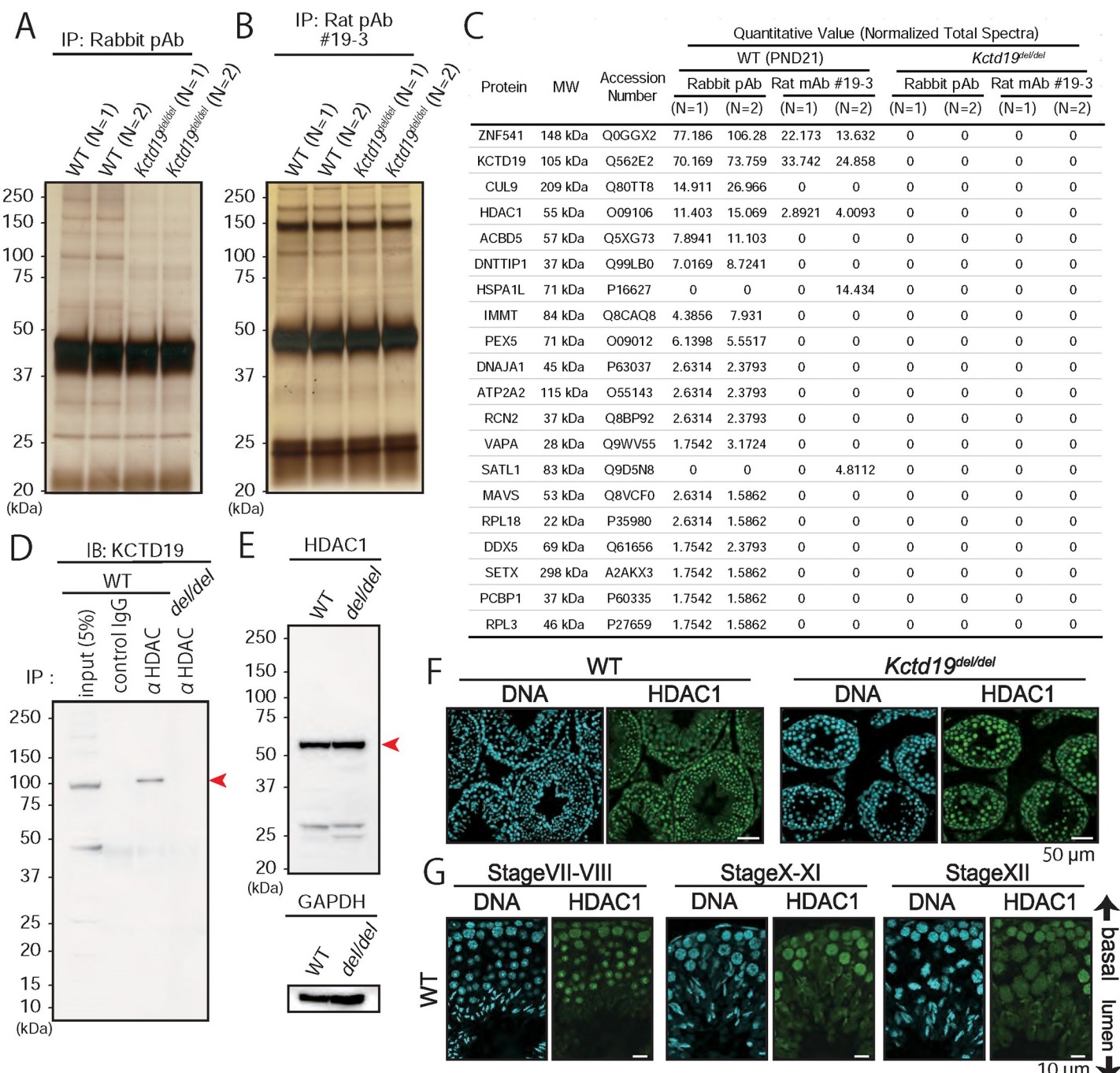

**Fig 5. IP-MS analysis with anti-KCTD19 antibody.** (A & B) Silver stain of eluted IP samples from rabbit pAb (A) and rat mAb #1 (B). Two juvenile WT mice and two adult *Kctd19<sup>del/del</sup>* mice were used for each experiment. (C) The list of identified proteins by MS analysis. The quantitative value was calculated using Scaffold software. (D) Immunoprecipitation with an anti-HDAC1 antibody. For input sample, 50 μg of testis lysate was used. (E) Immunoblotting with an anti-HDAC1 antibody. An anti-GPADH antibody was used as a loading control. (F and G) Immunostaining with an anti-HDAC1 antibody. The seminiferous epithelium cycle was determined by cell position, nuclear morphology, and morphology of the acrosome stained with Alexa Flour 568-conjugated lectin PNA.

Finally, the chimeric analysis showed that *Zfp541* KO spermatocytes failed to exit the pachytene stage, unlike *Kctd19* KO spermatocytes which underwent apoptosis during the metaphase-anaphase transition. *Zfp541* KO spermatocytes failed XY chromosome synapsis,

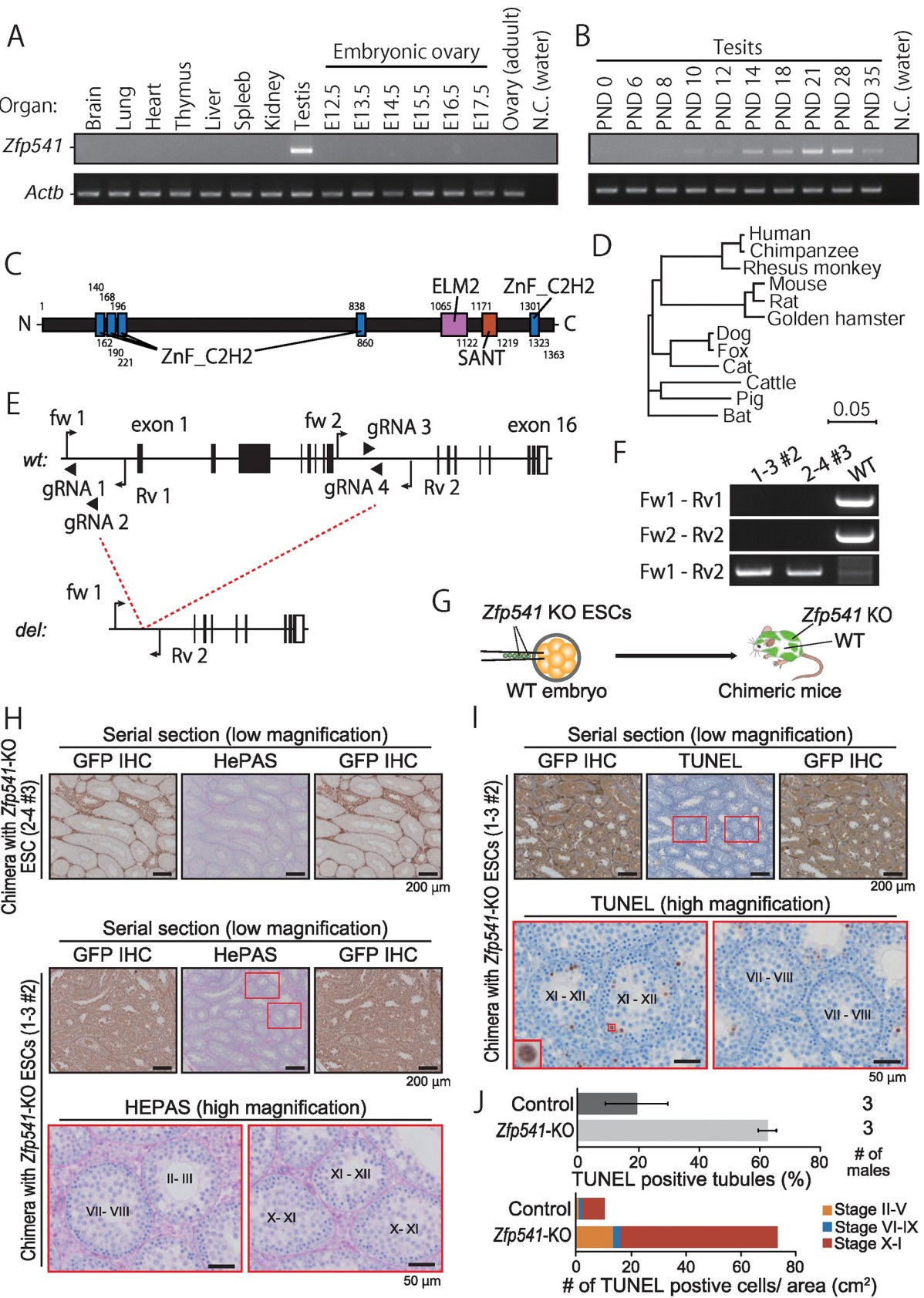

**Fig 6. Chimeric analysis of *Zfp541* KO spermatocytes.** (A) RT-PCR using multi-tissue cDNA. *Actb* was used as a loading control. (B) RT-PCR using postnatal testis cDNA. *Actb* was used as a loading control. (C) Schematic of ZFP541 protein structure. (D) Phylogenetic tree constructed by ClustalW with ZFP541 sequences of various mammals. (E) Gene map of *Zfp541*. Black and white boxes indicate coding and non-coding regions, respectively. Black arrows and arrowheads indicate primers for genotyping and gRNAs for genome editing, respectively. (F) Genotyping PCR with two primer sets in C for ES cell clones used in this study. (G) Schematic of chimeric mice production. ESC-derived cells were labeled with GFP fluorescence. (H) PAS staining of seminiferous tubules of chimeric mice. ESC-derived *Zfp541*-KO spermatocytes were identified by GFP-IHC of upper and lower serial sections. The seminiferous epithelial stages were roughly determined by the arrangement and nuclear morphology of the first layer of germ cells (spermatogonia and leptotene/zygotene spermatocytes). (I) TUNEL staining of seminiferous tubules of chimeric mice counterstained with hematoxylin. ESC-derived *Zfp541*-KO spermatocytes were identified by GFP-IHC of upper and lower serial sections. (J) Quantification of TUNEL staining. The seminiferous epithelial stages were roughly determined by the arrangement and nuclear morphology of the first layer of germ cells (spermatogonia and leptotene/zygotene spermatocytes). Control is *Kctd19^{we/del}* used in Fig 2E.

and γH2AX foci signals regained outside the XY body in the late pachytene stage, resulting in apoptosis. It is known that SPO11 and other DSB proteins persist on chromosomes into the pachytene stage, and engagement of homologous chromosomes is one mechanism for restraining DSB protein activity [42]. Therefore, precocious SYCP1 dissociation (desynapsis) or delays in other processes might cause the reactivation of DSB proteins remaining on chromosomes. Again, we acknowledge that these pachytene structural defects might be secondary effects or phenomena in dying cells.

In summary, our results showed that KCTD19 associates with ZFP541 and HDAC1 and are essential for meiosis. Further comparable studies will unveil the exact functions of KCTD19 and ZFP541, which will provide some insight into the molecular mechanism in male meiosis.

## Materials and methods

### Ethics statement

All animal experiments were approved by the Animal Care and Use Committee of the Research Institute for Microbial Diseases, Osaka University (#Biken-AP-H30-01). Animals were housed in a temperature-controlled environment with 12 h light cycles and free access to food and water. B6D2F1 (C57BL/6 × DBA2; Japan SLC, Shizuoka, Japan) mice and ICR (SLC) were used as embryo donors; B6D2F1 were used for mating and wild-type controls; C57BL6/N (SLC) mice were used to collect RNA for RT-PCR and cloning. Gene-manipulated mouse lines used in this study will be deposited at both the Riken BioResource Center (Riken BRC, Tsukuba, Japan) and the Center for Animal Resources and Development, Kumamoto University (CARD, Kumamoto, Japan). All lines are available through these centers.

### Egg collection

To prepare eggs for knockout mouse production, female mice were superovulated by injection of CARD HyperOva (0.1 mL, Kyudo, Saga, Japan) into the abdominal cavity of B6D2F1 females, followed by injection of human chorionic gonadotropin (hCG) (7.5 units, ASKA Pharmaceutical, Tokyo, Japan). Natural mating was done with B6D2F1 males 46~48 h after CARD HyperOva injection. After 19–21 h, cumulus-intact eggs were collected and treated with 0.33 mg/mL hyaluronidase (Wako, Osaka, Japan) for 5 min to remove cumulus cells for genome editing. Obtained eggs were cultured in KSOM medium at 37°C under 5% $CO_2$ until subsequent treatments.

### Generation of Kctd19 deletion and Kctd19-ΔPOZ/TAZ mice

*Kctd19* deletion mice and *Kctd19*-ΔPOZ/TAZ mice were generated by electroporation described previously [43,44]. Briefly, a gRNA solution was prepared by annealing two

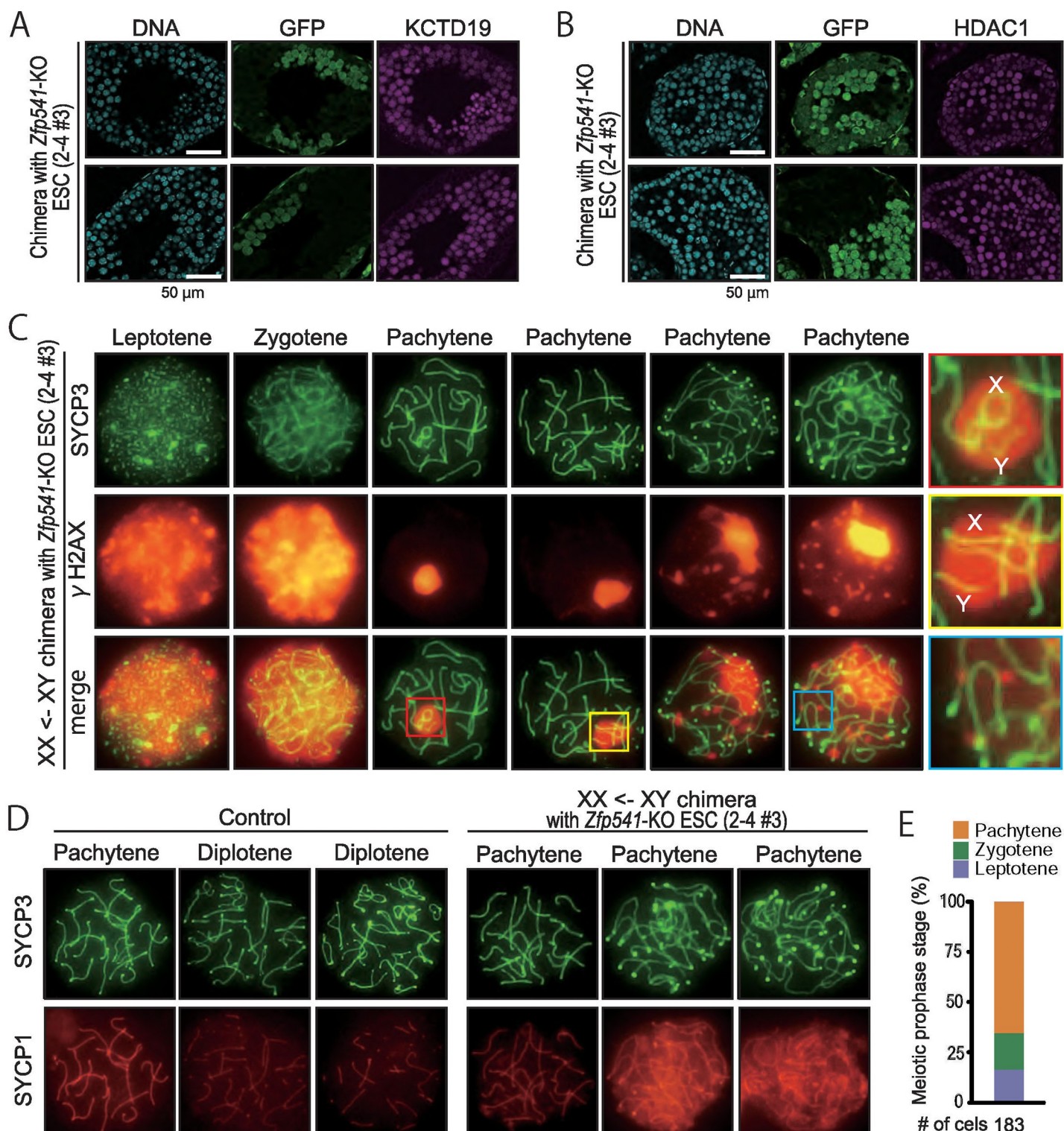

**Fig 7. Immunostaining analysis of *Zfp541* KO spermatocytes.** (A and B) Immunostaining of seminiferous tubules of chimeric mice with antibodies against KCTD19 (A) and HDAC1 (B). ES cell-derived *Zfp541*-KO spermatocytes were identified by GFP immunostaining. (C) Immunostaining of spread nuclei of prophase spermatocytes collected from XY/XX chimeric mice. Red, yellow, and blue boxes are magnified in the right panels. (D) Immunostaining of spread nuclei with anti-SYCP1 and anti-SYCP3 antibodies. *Kctd19^{we/del}* spermatocytes was used as control. (E) The percentage of cells in various meiotic prophase stages counted from immunostained spread nuclei samples.

tracrRNAs (Sigma-Aldrich, St. Louis, MO, USA) and crRNA (Sigma-Aldrich). The target genomic sequences are listed in S1 Table. Then, the gRNA solution and Cas9 nuclease solution (Thermo Fisher Scientific, Waltham, MA, USA) were mixed. The final concentrations of gRNA and Cas9 were as follows: for pronuclear injection, 20 ng/μL gRNA, and 100 ng/μL Cas9 nucleases. The obtained complex was electroporated into fertilized eggs using a NEPA21 electroporator (NEPA GENE, Chiba, Japan). The electroporated eggs were transplanted into the oviduct ampulla of pseudopregnant mice (ICR; 10 embryos per ampulla) on the following day. After 19 days, pups were delivered through Caesarean section and placed with foster mothers (ICR). To generate heterozygous mutant mice, F0 mice were mated with WT B6D2F1. Mouse colonies with a 9612 bp deletion and a 2172 bp deletion were maintained by sibling mating and used for the phenotype analysis of *Kctd19* deletion and *Kctd19*-ΔPOZ, respectively. The genotyping primers (GeneDesign, Osaka, Japan) and amplification conditions are available in S1 Table.

## Generation of 3xFLAG-Kctd19 and Kctd19-3xHA transgenic mice

The mouse *Kctd19* cDNA (ENSMUST00000167294.7) was tagged with 3xFLAG or 3xHA with a rabbit polyA signal inserted under the control of the mouse Clgn promoter. After linearization, an equal amount of the DNA constructs (2.16 ng/μL; 0.54 ng/μL/kbp) were mixed and injected into the pronucleus of fertilized eggs. The injected eggs were transplanted into the oviduct ampulla of pseudopregnant mice (ICR; 10 embryos per ampulla) the following day. After 19 days, pups were delivered through Caesarean section and placed with foster mothers (ICR). Offspring carrying both the 3xFLAG tag-*Kctd19* and *Kctd19*-3xHA transgenes and mice carrying only 3xHA tag-*Kctd19* transgene were used in this study. The genotyping primers (Gene-Design) are available in S1 Table.

## Generation of Zfp541 KO ES cells and chimeric mice

*Zfp541* KO embryonic stem (ES) cells were generated using methods previously described (5). Briefly, EGR-G01 ES cells were transfected with two pX459 plasmids (Addgene plasmid #62988) with the target sequences (S1 Table), and colonies were selected after transient puromycine selection. ES cells with normal karyotypes were injected into ICR embryos and chimeric blastocysts were transferred into the uteri of pseudopregnant females to produce chimeric offspring. Chimeric males with high ES cell contribution were used for experiments.

## Cell lines

EGR-G01 ES cells were generated in the Ikawa Lab [28] and cultured in KnockOut DMEM (108297–018, Thermo Fisher Scientific) supplemented with 1% Penicillin-Streptomycin- Glutamine, 55 μM 2-mercaptoethanol, 1% Non-Essential Amino Acid Solution (11140–050, Thermo Fisher Scientific), 1% Sodium Pyruvate (11360–070, Thermo Fisher Scientific), 30 μM Adenosine (A4036, Sigma- Aldrich, St. Louis, MO, USA), 30 μM Guanosine (G6264, Sigma-Aldrich), 30 μM Cytidine (C4654, Sigma-Aldrich), 30 μM Uridine (U3003, Sigma-Aldrich), 10 μM Thymidine (T1895, Sigma-Aldrich), 100 U/ml mouse LIF, and 20% FCS (51650–500, Biowest, Nuaillé, France).

## Bacterial strains

*Escherichia coli* (*E. coli*) strain DH5α (Toyobo, Osaka, Japan) and BL21(de3) pLysS (C606003, ThermoFisher Scientific) were used for DNA cloning and protein expression, respectively. *E.*

*coli* cells were grown in LB or 2×YT medium containing 100 mg/L ampicillin and were transformed or cloned using standard methods.

## Production of antibodies against KCTD19

A polyclonal antibody against mouse KCTD19 was generated by immunizing rabbits with the synthetic peptide KRAITLKDWGKQRPKDRES corresponding to amino acids 747–765 of mouse KCTD19 (NP_808459.1). For monoclonal antibody production, the DNA encoding mouse KCTD19 (residue 654–793 aa, NP_808459.1) was inserted into pGEX6p-1 (GE healthcare), and the expression vector was transformed into *E. coli* strain BL21 (de3) pLysS (C606003, Thermo Fisher Scientific). GST-KCTD19 was purified using Glutathione Sepharose 4B (GE Healthcare). The purified KCTD19 protein with a complete adjuvant was injected into female rats. After 17 days of injection, lymphocytes were collected from iliac lymph nodes and hybridomas were generated [45–46]. The cell clones were screened by limited dilution.

## Sequence comparison analysis

Amino acid sequences of KCTD19 and ZFP541 were obtained from the NCBI Entrez Protein database. Clustal W2.1 was used for multiple sequence alignment [21].

## RT-PCR

Using TRIzol reagent (15596–018, ThermoFisher Scientific), total RNA was isolated from multiple adult tissues of C57BL6/N mice, testes ranging from 1 to 35-day-old mice, and embryonic ovaries of PND 11.5–19.5. cDNAs were prepared using SuperScript IV Reverse Transcriptase (180–90050, ThermoFisher Scientific) following the manufacturer's instructions. Polymerase chain reaction (PCR) was performed using KOD Fx neo (KFX-201, TOYOBO, Osaka, Japan). The primers (GeneDesign) and amplification conditions for each gene are summarized in S1 Table.

## Genotype analysis

PCR was performed using KOD FX neo (KFX-201, TOYOBO). The primers (GeneDesign) and amplification conditions for each gene are summarized in S1 Table. PCR products were purified using a Wizard SV Gel and PCR Clean-Up System (Promega, Madison, WI, USA) kit, and Sanger sequenced was done using sequencing primers listed in S1 Table.

## Fertility analysis of KO mice

To examine fertility, sexually mature male mice were housed with wild-type females (B6DF1) for at least three months. Both plug and pup numbers were recorded at approximately 10 AM to determine the number of copulations and litter size. Numerical data is available in S4 Table.

## Immunoblotting

Proteins from testis were extracted using NP40 lysis buffer [50mM Tris-HCl (pH 7.5), 150 mM NaCl, 0.5% NP-40, 10% Glycerol, protease inhibitors]. Proteins were separated by SDS-PAGE under reducing conditions and transferred to polyvinylidene fluoride (PVDF) membrane using the Trans Blot Turbo system (BioRad, Munich, Germany). After blocking with 10% skim milk (232100, Becton Dickinson, Cockeysville, MD, USA), the membrane was incubated with primary antibody overnight at 4˚C, and then incubated with HRP-conjugated secondary antibody for 1 h at room temperature. Chemiluminescence was detected by ECL Prime Western Blotting Detection Reagents (RPN2232, GE Healthcare, Chicago, IL, USA)

using the Image Quant LAS 4000 mini (GE Healthcare). The antibodies used in this study are listed in S2 Table.

## Morphological and histological analysis of testis

To observe testis gross morphology and measure testicular weight, 11–12 week-old male mice were euthanized after measuring their body weight. The whole testis was observed using BX50 and SZX7 (Olympus, Tokyo, Japan) microscopes. For histological analysis, testes were fixed with Bouin's fixative (16045–1, Polysciences, Warrington, PA, USA) at 4°C O/N, dehydrated in increasing ethanol concentrations and 100% xylene, embedded in paraffin, and sectioned (5 μm). The paraffin sections were hydrated with Xylene and decreasing ethanol concentrations and treated with 1% periodic acid (26605–32, Nacalai Tesque, Kyoto, Japan) for 10 min, treated with Schiff's reagent (193–08445, Wako) for 20 min, counterstained with Mayer's hematoxylin solution (131–09665, Wako) for 3 min, dehydrated in increasing ethanol concentrations, and finally mounted with Permount (SP15-100-1, Ferma, Tokyo, Japan). The sections were observed using a BX53 (Olympus) microscope.

## Apoptosis detection in testicular section

TdT-mediated dUTP nick end labeling (TUNEL) staining was carried out with In Situ Apoptosis Detection Kit (MK500, Takara Bio Inc., Shiga, Japan), according to the manufacturer's instruction. Briefly, testes were fixed with Bouin's fixative, embedded in paraffin, and sectioned (5 μm). After paraffin removal, the slides were boiled in citrate buffer (pH 6.0; 1:100; ab93678, abcam, Cambridge, UK) for 10 min and incubated in 3% $H_2O_2$ at room temperature for 5 min for endogenous peroxidase inactivation, followed by a labeling reaction with TdT enzyme and FITC-conjugated dUTP at 37°C for 1 h.

For chromogenic detection of apoptosis, the sections were incubated with HRP-conjugated anti-FITC antibody at 37°C for 30 min. The section was then incubated in ImmPACT DAB (SK-4105, Vector Laboratories, Burlingame, CA, USA) working solution, counterstained with Mayer's hematoxylin solution for 3 min, dehydrated in increasing ethanol concentrations, and finally mounted with Permount. The sections were observed using a BX53 (Olympus) microscope. Numerical data is available in S4 Table.

## Immunostaining of testes

Testes were fixed in 4% paraformaldehyde (PFA) overnight at 4°C, followed by dehydration in increasing ethanol concentrations and 100% xylene, embedded in paraffin, and sectioned (5 μm). After paraffin removal, the slides were boiled in pH 6.0 citrate buffer for 10 min, blocked and permeabilized with 10% goat serum and 0.1% TritonX-100 for 20 min in PBS, and incubated with primary antibody overnight at 4°C or 1 h at room temperature in blocking solution; 1 h incubation was performed when using rat monoclonal anti-KCTD19 antibody. After incubation with Alexa Flour 488/546-conjugated secondary antibody (1:200) at room temperature for 1 h, samples are counterstained with Hoechst 33342 (1:2000; H3570, Thermo Fisher Scientific) and mounted with Immu-Mount (9990402, Thermo Fisher Scientific). The antibodies used in this study are listed in S2 Table.

Seminiferous tubule stages were identified based on the morphological characteristics of the germ cell nuclei and acrosome staining with Alexa Flour 488/568-conjugated lectin PNA (L21409/L32458, Thermo Fisher Scientific). The sections were observed using a BX53 (Olympus) microscope and a Nikon Eclipse Ti microscope connected to a Nikon C2 confocal module (Nikon, Tokyo, Japan). Fluorescent images were false-colored and cropped using ImageJ software.

## Immunostaining of surface chromosome spreads

Spread nuclei from spermatocytes were prepared as described [47] with slight modification. In brief, seminiferous tubules were unraveled using forceps in ice-cold DMEM (11995065, Thermo Fisher Scientific) and incubated in 1 mg/mL collagenase (C5138, Sigma-Aldrich) in DMEM (20 mL) at 37°C for 15 min. After 3 washes with DMEM, the tubules were transferred to 20 mL trypsin/DNaseI medium [0.025 w/v% trypsin, 0.01 w/v% EDTA, 10U DNase in DMEM] and incubated at 37°C for 10 min. After adding 5 mL of heat-inactivated FCS and pipetting, the solution was filtered through a 59 μm mesh (N-N0270T, NBC Meshtec inc., Tokyo, Japan) to remove tubule debris. The collected testicular cells were resuspended in hypotonic solution [100 mM sucrose] and 10 μL of the suspension was dropped onto a glass slide with 100 μL of fixative solution [100 μL of 1% PFA, 0.1% (v/v) Triton X-100]. The slides were then air-dried and washed with PBS containing 0.4% Photo-Flo 200 (1464510, Kodak Alaris, NY, USA) or frozen for longer storage at -80°C.

The spread samples were blocked with 10% goat serum in PBS and then incubated with primary antibodies overnight at 4°C in blocking solution. After incubation with Alexa Flour 488/546-conjugated secondary antibody (1:200) at room temperature for 1 h, samples are counterstained with Hoechst 33342 and mounted with Immu-Mount. The samples were observed using a BX53 (Olympus) microscope. Numerical data is available in S4 Table.

## Giemsa staining of metaphase I chromosome spreads

For preparing metaphase chromosome spreads, seminiferous tubules were unraveled using forceps in ice-cold PBS and transferred to a 1.5-mL tube with 1 mL of accutase (12679–54, Nacalai Tesque), followed by clipping the tubules, and a 5 min incubation at room temperature. After filtration with a 59 μm mesh and centrifugation, the cells were resuspended in 8 mL of hypotonic solution [1% sodium citrate] and incubated for 5 min at room temperature. The suspension was centrifuged and 7 mL of supernatant was aspirated. The cells were then resuspended in the remaining 1 mL of supernatant and 7 mL of Carnoy's Fixative (75% Methanol, 25% Acetic Acid) were added gradually while shaking. After 2 washes with Carnoy's Fixative, the cells were resuspended in ~ 0.5 mL of Carnoy's Fixative and dropped onto a wet glass slide. The slide was stained with Giemsa Stain Solution (079–04391, wako) and observed using a BX53 (Olympus) microscope.

## Immunostaining of metaphase I cells

For cytological analysis of metaphase I cells, seminiferous tubule squashes were performed as previously described [48]. In brief, seminiferous tubules were incubated in fix/lysis solution [0.1% TritonX-100, 0.8% PFA in PBS] at room temperature for 5 min. Tubule bunches were then put on glass slides with 100 μL of fix/lysis solution, minced into 1.0 ~ 3.0 mm segments with forceps, and arranged so that no tubule segment overlapped. After removing the excess amount of fix/lysis solution, a coverslip and pressure was applied to disperse cells, followed by flash freezing in liquid nitrogen for 15 sec, and removing the coverslip with forceps and a needle. For longer storage, the glass slides were kept at -80°C with the coverslip.

The slides were blocked and permeabilized in 10% goat serum and 0.1% Triton X-100 for 20 min in PBS, and incubated with primary antibody overnight at 4°C. After incubation with Alexa Flour 488/546-conjugated secondary antibody (1:200) at room temperature for 1 h, samples are counterstained with Hoechst 33342 (1:2000) and mounted with Immu-Mount. Z-stack images were taken using a BZ-X700 (Kyence, Osaka, Japan) microscope and stacked using ImageJ software. The antibodies used in this study are listed in S2 Table.

## Immunoprecipitation and mass spectrometry analysis

Proteins from testis were extracted using NP40 lysis buffer [50 mM Tris-HCl (pH7.5), 150 mM NaCl, 0.5% NP-40, 10% Glycerol, protease inhibitor]. Protein lysates were mixed with Dynabeads Protein G (Thermo)-conjugated with 2.0 μg of antibody. The immune complexes were incubated for 1 h at 4˚C and washed 3 times with NP40 lysis buffer. Co-immunoprecipitated products were then eluted with 18 μL of 100 mM Gly-HCl (pH 2.5) and neutralized with 2 μL of 1 M Tris. The antibodies used in this study are listed in S2 Table. Half of the eluted amount was subjected to SDS-PAGE and silver staining (06865–81, Nacalai Tesque). The remaining amount was subjected to mass spectrometry (MS) analysis.

The proteins were reduced with 10 mM dithiothreitol (DTT), followed by alkylation with 55 mM iodoacetamide, and digested by treatment with trypsin and purified with a C18 tip (GL-Science, Tokyo, Japan). The resultant peptides were subjected to nanocapillary reversed-phase LC-MS/MS analysis using a C18 column (25 cm × 75 um, 1.6 μm; IonOpticks, Victoria, Australia) on a nanoLC system (Bruker Daltoniks, Bremen, Germany) connected to a tims TOF Pro mass spectrometer (Bruker Daltoniks) and a modified nano-electrospray ion source (CaptiveSpray; Bruker Daltoniks). The mobile phase consisted of water containing 0.1% formic acid (solvent A) and acetonitrile containing 0.1% formic acid (solvent B). Linear gradient elution was carried out from 2% to 35% solvent B for 18 min at a flow rate of 400 nL/min. The ion spray voltage was set at 1.6 kV in the positive ion mode. Ions were collected in the trapped ion mobility spectrometry (TIMS) device over 100 ms and MS and MS/MS data were acquired over an *m/z* range of 100–1,700. During the collection of MS/MS data, the TIMS cycle was adjusted to 1.1 s and included 1 MS plus 10 parallel accumulation serial fragmentation (PASEF)-MS/MS scans, each containing on average 12 MS/MS spectra (>100 Hz), and nitrogen gas was used as the collision gas.

The resulting data were processed using DataAnalysis version 5.1 (Bruker Daltoniks), and proteins were identified using MASCOT version 2.6.2 (Matrix Science, London, UK) against the SwissProt database. Quantitative value (available in S3 Table) and fold exchange were calculated by Scaffold4 (Proteome Software, Portland, OR, USA) for MS/MS-based proteomic studies.

## Chimeric analysis

For distinguishing ESC-derived germ cells, GFP was stained by immunofluorescence or immunohistochemistry. The antibodies used in this study are listed in S2 Table.

## Supporting information

**S1 Fig. Sequence comparison of KCTD19 in various mammals, related to Fig 1D.** Prortein sequence comparison of KCTD19 in cattle (NP_001098862.1), pig (XP_003126977.2), dog (XP_022275030.1), fox (XP_025867456.1), cat (XP_023101865.1), bat (XP_027998908.1), human (NP_001094385.1), chimpanzee (XP_523391.2), rhesus monkey (XP_014981866.1), mouse (NP_808459.1), rat (NP_001292128.1), and golden hamster (XP_021086458.1). (TIF)

**S2 Fig. Production of *Kctd19*-ΔBTB mice and fertility analysis.** (A) Gene map of *Kctd19*. Black and white boxes indicate coding and non-coding regions, respectively. Black arrows and arrowheads indicate primers for genotyping and gRNAs for genome editing, respectively. (B) An example of genotyping PCR with two primer sets shown in S2A. (C) DNA sequencing verifies the deletion. (D) RT-PCR using testis cDNA obtained from WT and *ΔBTB/ΔBTB* mice. *Actb* was used as a loading control. (E) Immunoblotting using testis lysates obtained from WT,

*del*/*del*, and *ΔBTB*/*ΔBTB* mice. (F) PAS staining of seminiferous tubules of adult mice. The seminiferous epithelium cycle was determined by germ cell position and nuclear morphology. (TIF)

**S3 Fig. Chromosome spreads of *Kctd19*<sup>*del*/*del*</sup> spermatocytes (PND20), related to Fig 3A and 3B.** (A) Immunostaining of spread nuclei from prophase spermatocytes collected from juvenile mice (PND20). (B) The percentage of each meiotic prophase stage present is determined by immunostained spread nuclei samples. (TIF)

**S4 Fig. Sequence comparison of ZFP541 in various mammals, related to Fig 6D.** Prortein sequence comparison of ZFP541 proteins from various mammals: cattle (XP_015313711.2), pig (XP_020950303.1), dog (XP_005616437.1), fox (XP_025869832.1), cat (XP_023100994.1), bat (XP_008152641.1), human (NP_001264004.1), chimpanzee (XP_016791837.1), rhesus monkey (XP_014979842.2), mouse (NP_001092747.1), rat (NP_001100928.2), and golden hamster (XP_021078928.1). (TIF)

**S5 Fig. Production of XY/XX chimeric mice and their features, related to Fig 7C–7E.** (A) Schematic of XY/XX chimeric mice production. XX prospermatogonia are eliminated around PND2. (B) Testis sections from chimeric mice. ES cell-derived cells were labeled with GFP fluorescence. Asterisk indicates depleted tubules. (TIF)

**S1 Table. Primers and gRNAs used in this study.** (XLSX)

**S2 Table. Antibodies used in this study.** (XLSX)

**S3 Table. The quantitative value of mass spectrometry analysis.** (XLSX)

**S4 Table. Numerical data that underlies graphs.** (XLSX)

## Acknowledgments

We would like to thank Eri Hosoyamada and Mei Koyama for their technical assistance and Dr. Julio M. Castaneda for the critical reading of the manuscript.

## Author Contributions

**Conceptualization:** Seiya Oura, Masahito Ikawa.

**Data curation:** Seiya Oura, Masahito Ikawa.

**Formal analysis:** Seiya Oura.

**Funding acquisition:** Seiya Oura, Kei-ichiro Ishiguro, Masahito Ikawa.

**Investigation:** Seiya Oura.

**Methodology:** Seiya Oura.

**Project administration:** Masahito Ikawa.

**Resources:** Seiya Oura, Takayuki Koyano, Chisato Kodera, Yuki Horisawa-Takada, Makoto Matsuyama, Kei-ichiro Ishiguro, Masahito Ikawa.

**Software:** Seiya Oura.

**Supervision:** Masahito Ikawa.

**Validation:** Seiya Oura.

**Visualization:** Seiya Oura.

**Writing – original draft:** Seiya Oura, Masahito Ikawa.

**Writing – review & editing:** Seiya Oura, Masahito Ikawa.

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
