## [Decision Letter · Decision Letter 0]

11 Mar 2021

Dear Masa,

Thank you very much for submitting your Research Article entitled 'KCTD19 associates with ZFP541 and HDAC1 and is required for meiotic exit in male mice' to PLOS Genetics.

The manuscript was fully evaluated at the editorial level and by independent peer reviewers. The reviewers appreciated the attention to an important topic but identified some concerns that we ask you address in a revised manuscript. These are relatively minor, and many require mostly textual alterations.

We therefore ask you to modify the manuscript according to the review recommendations. Your revisions should address the specific points made by each reviewer.

[LINK]

Yours sincerely,

Paula E. Cohen

Associate Editor

PLOS Genetics

Gregory P. Copenhaver

Editor-in-Chief

PLOS Genetics

Reviewer's Responses to Questions

**Comments to the Authors:**

Reviewer #1: This manuscript reports a detailed study of KCTD19 function in mouse meiosis. It describes the expression and subcellular localization pattern of KCTD19 throughout spermatogenesis. Kctd19 knockout mice were generated by CRISPR/Cas9 approach. They find that Kctd19 mutant spermatocytes progress through meiotic prophase I with no defects in chromosomal synapsis and meiotic recombination but are arrested at the metaphase I stage. The Kctd19 KO phenotype can be rescued by transgenes expressing HA or FLAG-tagged KCTD19. IP/mass spec analysis of KCTD19-containing protein complexes identified ZFP541 and HDAC1. Chimeric mutant analysis showed that ZFP541 is also required for meiosis and that Zfp541-deficient spermatocytes fail to progress to the diplotene stage. The data support their conclusions. The manuscript is well-written. The literature review is balanced and thorough. This study contributes interesting findings to the field of meiosis.

Major concerns:

1) Regarding the term “meiotic exit”, I suggest that the term “meiotic exit” not be used. The Kctd19 mutant spermatocytes are arrested at metaphase I. Zfp541 mutant spermatocytes are arrested at the pachytene stage. Clearly, both are essential for meiosis (or meiotic progression). The mutant spermatocytes have yet to reach meiosis II. Meiotic exit gives the impression that the mutant germ cells cannot exit into the post-meiotic round spermatid stage. I think that the term “meiotic exit” is confusing and not appropriate. My suggestion is to change “meiotic exit” to “meiosis” or “meiotic progression” throughout the manuscript.

2) Chimeric (XX/XY) mice were generated by injection of Zfp541 KO ES cells into blastocysts to study the requirement of Zfp541 function. This approach lacks an explanation/rationale. Why not injecting Zfp541+/- (Het) ES cells to generate chimera mice for germline transmission? Is it because Zfp541+/- (Het) ES cells were never obtained? If this were the case, a few sentences of explanation/description would be very useful.

Minor:

1) Line 35, change “that consisting…” to “that consists…”

2) Line 202, change “anit-KCT19” to “anti-KCTD19”.

Reviewer #2: REVIEW: KCTD19 ASSOCIATES WITH ZFP541 AND HDAC1 AND IS REQUIRED FOR MEIOTIC EXIT IN MALE MICE

SUMMARY

Here, Oura et al. screened evolutionarily conserved and reproductive-tract enriched genes using the CRISPR/Cas9 system and identified potassium channel tetramerization domain containing 19 (Kctd19) as an essential regulator of meiosis. Oura et al. generated a Kctd19 KO mouse line, demonstrated the loss of KCTD19 expression using polyclonal and monoclonal antibodies, and show that these mice are unable to produce litters with wildtype females (fig. 1). Histological analysis of the Kctd19 null male testes also revealed significantly smaller testis-to-body weight ratios, a loss of normal testicular architecture, and an increase in the number of cells undergoing apoptosis, implicating Kctd19 in sperm production in male mice (fig. 2). These spermatocytes also could not complete meiosis and accumulated in the seminiferous tubules (fig. 2). Immunostaining of prophase spermatocytes with antibodies against the synaptonemal complex and spindle apparatus show that Kctd19 null spermatocytes experience a significant increase in chromosome misalignment and the formation of SYCP3 polycomplexes, indicating that KCTD19 may be important for metaphase I organization (fig. 3). In fig. 4, the authors were able to rescue the Kctd19 null phenotype with an epitope-tagged Kctd19 transgene. In fig. 5, Oura et al. show that ZFP541 and HDAC1 are putative interactors with KCTD19 via IP-MS. Overall, the authors claim that KCTD19 associates with ZFP541 after the late pachytene stage, in addition to HDAC1, though the interaction timing of both complexes remains unknown.

One major strength of this paper is the interesting finding that KCTD19 likely associates with ZFP541 after the late pachytene stage and that KCTD19 is essential for the metaphase-anaphase transition. However, the following suggestions may improve the manuscript:

MAJOR CONCERNS

1. One major concern of this paper is that Oura et al. recognize that Choi et al. already demonstrated that KCTD19 complexes with ZFP541, which was the main conclusion of this paper by Oura et al. This suggests that the findings in this paper have already been reported and more follow-up experiments are needed to offer novel results.

2. The antibodies used in Fig. 1h, 4c, 4e, 5a, and 5b do not appear to be specific. This is particularly concerning since these antibodies were used for the IP-MS experiment.

MINOR CONCERNS

1. The labeling system of the KO lines, transgenic lines, and the antibodies used was not clear or easy to follow.

2. The chromosome spreads presented in fig. 3 are not spread enough to allow the reader to look at SC morphology. It would also be helpful to have the yH2AX signal overlayed with the SYCP3 channel to investigate sex body formation.

3. There are grammatical and syntax errors throughout this manuscript which made it hard to understand some of the descriptions.

Reviewer #3: In the manuscript by Oura et al. titled “KCTD19 associates with ZFP541 and HDAC1 and is required for meiotic exit in male mice”, the authors create Kctd19 knockout mice and show that KCTD19 loss leads to infertility in male mice. Meiotic cells lacking KCTD19 do not progress beyond the metaphase stage and this phenotype is rescued by a tagged Kctd19 transgene. KCTD19 has been previously shown to interact with ZFP541 and HDAC1 and the authors confirm this through IP-MS of KCTD19. They also delete Zfp541 in ES cells, create chimeric mice and show that meiotic cells derived from the targeted ES cells have DNA damage present during the pachytene stage and do not develop past the pachytene stage.

While the mechanisms of action of KCTD19 and ZFP541 are not investigated, the authors present the initial characterization of new meiotic proteins and their mutants. Few major and minor comments are listed below:

Major comments:

The title states that KCTD19 is required for meiotic exit. And the abstract states that KCTD19 and ZFP541 are essential for meiotic exit. The phenotype observed by the authors in mice lacking KCTD19 is metaphase arrest and lack of round spermatids. The phenotype observed in mice lacking ZFP541 is depletion of diplotene cells, presence of gamma-H2AX during late prophase and absence of round spermatids. Although these phenotypes occur late during prophase, they do not necessarily suggest a role for KCTD19 and ZFP541 in “meiotic exit”. And there is no data provided in the manuscript to suggest that. I suggest modifying the text and title to better represent what is shown, or providing additional background and discussion detailing why the authors conclude this role for KCTD19 and ZFP541.

There is another instance where the language used is too strong and not supported by the data: in the abstract the authors state that KCTD19 interacts with HDAC1 and that this indicated that KCTD19 is involved in chromatin modification. While this may be the case, this statement should be modified or moved to the discussion section given there is no additional data presented to support a role for KCTD19 in chromatin modification.

The images in Figure 2(C) are difficult to see. The authors claim that metaphase cells are present in mutants at stage XII and remain present at stages I-II. The authors should annotate the metaphase cells with arrows or by circling them. And provide a magnified inset showing an example of the metaphase cells that they have annotated.

The same applies to Figure 2(D). It’s impossible to see which cells are TUNEL positive by the image shown. An example of the TUNEL positive cells should be shown as a magnified inset. And TUNEL staining should be quantified.

In Figure 6(F), cell types should be labeled to help interpretation of the data.

The authors state that spermatocytes lacking ZFP541 are eliminated prior to reaching metaphase and refer to Figure 6(G). It is impossible to interpret the images shown and the reader simply has to go by what is stated in the text. The images should clearly show which cells are both GFP positive and TUNEL positive and what stage are they in. Insets should be provided to clarify this. And TUNEL staining should be quantified.

Minor comments:

Since the antibodies used to detect KCTD19 are C-terminally located, can the authors exclude the possibility that the N-terminal region containing the BTB domain is expressed and stable in the del allele? This can be easily tested if there is an available antibody against the N-terminus.

In figures plotting testis weights, the same units (either mg or g) should be used for both the testis and body weights.

In Figure 2(G) it would be useful to have cell types annotated (e.g. pachytene, round spermatid etc.).

In figure 5(F) the DNA and HDAC1 images don’t match for stage VII-VIII. Please correct.

A figure showing protein domain structure of ZFP541 within Figure 6 would be nice.

The absence of nucleus-wide gamma-H2AX in early pachytene but presence in late pachytene in cells lacking ZFP541 is interesting. Does this imply that new damage is being introduced at late pachytene? This phenotype is not discussed. How do the authors interpret this?

There are several typos in figures: e.g. the words metaphase (3D) and pachytene (6J) are misspelled. And typos are present in figure legends: e.g. legend of Figure 5 (D) is incorrect, there is a typo in legend of Figure 5 (E) and S2(E), and S2 refers delta-POZ mice whereas figure labels state delta-BTB mice. The title of the first results section also has a typo.

**Have all data underlying the figures and results presented in the manuscript been provided?**

Reviewer #1: Yes

Reviewer #2: Yes

Reviewer #3: Yes

PLOS authors have the option to publish the peer review history of their article (what does this mean?). If published, this will include your full peer review and any attached files.

Reviewer #1: **Yes: **Jeremy Wang

Reviewer #2: No

Reviewer #3: No

---

## [Editor Report · Decision Letter 1]

5 Apr 2021

Dear Masa,

We are pleased to inform you that your manuscript entitled "KCTD19 and its associating protein ZFP541 are independently essential for meiosis in male mice" has been editorially accepted for publication in PLOS Genetics. Congratulations!

Warm wishes,

Paula E. Cohen

Associate Editor

PLOS Genetics

Gregory P. Copenhaver

Editor-in-Chief

PLOS Genetics

Comments from the reviewers (if applicable):

**Data Deposition**

http://datadryad.org/submit?journalID=pgenetics&manu=PGENETICS-D-21-00160R1

**Press Queries**

---

## [Editor Report · Acceptance letter]

26 Apr 2021

PGENETICS-D-21-00160R1 

KCTD19 and its associated protein ZFP541 are independently essential for meiosis in male mice  

Dear Dr Ikawa, 

We are pleased to inform you that your manuscript entitled "KCTD19 and its associated protein ZFP541 are independently essential for meiosis in male mice " has been formally accepted for publication in PLOS Genetics! Your manuscript is now with our production department and you will be notified of the publication date in due course.

With kind regards,

Katalin Szabo

PLOS Genetics

On behalf of:
